Computational bioacoustics with deep learning: a review and roadmap

http://orcid.org/0000-0001-8068-3769 Stowell Dan 1 2 d.stowell@tilburguniversity.edu
1 Department of Cognitive Science and Artificial Intelligence, Tilburg University , Tilburg , The Netherlands
2 Naturalis Biodiversity Center , Leiden , The Netherlands
Shang Yilun
Electronic publication date: 2022 Mar 21
Publication date: 2022
Volume: 10
Electronic Location ID: e13152
Received 2021 Dec 22; Accepted 2022 Mar 1
Copyright: © 2022 Stowell
Copyright year: 2022
Copyright holder: Stowell
License: This is an open access article distributed under the terms of the Creative Commons Attribution License, which permits unrestricted use, distribution, reproduction and adaptation in any medium and for any purpose provided that it is properly attributed. For attribution, the original author(s), title, publication source (PeerJ) and either DOI or URL of the article must be cited.
License URL: https://creativecommons.org/licenses/by/4.0/

Keywords: Acoustics, Deep learning, Bioacoustics, Animal vocal behaviour, Sound, Machine learning, Passive acoustic monitoring

Funding: The author received no funding for this work.

==============================
Animal vocalisations and natural soundscapes are fascinating objects of study, and contain valuable evidence about animal behaviours, populations and ecosystems. They are studied in bioacoustics and ecoacoustics, with signal processing and analysis an important component. Computational bioacoustics has accelerated in recent decades due to the growth of affordable digital sound recording devices, and to huge progress in informatics such as big data, signal processing and machine learning. Methods are inherited from the wider field of deep learning, including speech and image processing. However, the tasks, demands and data characteristics are often different from those addressed in speech or music analysis. There remain unsolved problems, and tasks for which evidence is surely present in many acoustic signals, but not yet realised. In this paper I perform a review of the state of the art in deep learning for computational bioacoustics, aiming to clarify key concepts and identify and analyse knowledge gaps. Based on this, I offer a subjective but principled roadmap for computational bioacoustics with deep learning: topics that the community should aim to address, in order to make the most of future developments in AI and informatics, and to use audio data in answering zoological and ecological questions.

Introduction

Bioacoustics—the study of animal sound—offers a fascinating window into animal behaviour, and also a valuable evidence source for monitoring biodiversity (Marler & Slabbekoorn, 2004; Laiolo, 2010; Marques et al., 2012; Brown & Riede, 2017). Bioacoustics has long benefited from computational analysis methods including signal processing, data mining and machine learning (Towsey et al., 2012; Ganchev, 2017). Within machine learning, deep learning (DL) has recently revolutionised many computational disciplines: early innovations, motivated by the general aims of artificial intelligence (AI) and developed for image or text processing, have cascaded through to many other fields (LeCun, Bengio & Hinton, 2015; Goodfellow, Bengio & Courville, 2016). This includes audio domains such as automatic speech recognition and music informatics (Abeßer, 2020; Manilow, Seetharman & Salamon, 2020).

Computational bioacoustics is now also benefiting from the power of DL to solve and automate problems that were previously considered intractable. This is both enabled and demanded by the twenty-first century data deluge: digital recording devices, data storage and sharing have become dramatically more widely available, and affordable for large-scale bioacoustic monitoring, including continuous audio capture (Ranft, 2004; Roe et al., 2021; Webster & Budney, 2017; Roch et al., 2017). The resulting deluge of audio data means that a common bottleneck is the lack of person-time for trained analysts, heightening the importance of methods that can automate large parts of the workflow, such as machine learning.

The revolution in this field is real, but it is recent: reviews and textbooks as recently as 2017 did not give much emphasis to DL as a tool, even when focussing on machine learning for bioacoustics (Ganchev, 2017; Stowell, 2018). Mercado & Sturdy (2017) reviewed the ways in which artificial neural networks (hereafter, neural networks or NNs) had been used by bioacoustics researchers; however, that review concerns the pre-deep-learning era of neural networks, which has some foundational aspects in common but many important differences, both conceptual and practical.

Many in bioacoustics are now grappling with deep learning, and there is much interesting work which uses and adapts DL for the specific requirements of bioacoustic analysis. Yet the field is immature and there are few reference works. This review aims to provide an overview of the emerging field of deep learning for computational bioacoustics, reviewing the state of the art and clarifying key concepts. A particular goal of this review is to identify knowledge gaps and under-explored topics that could be addressed by research in the next few years. Hence, after stating the survey methodology, I summarise the current state of the art, outlining standard good practice and the tasks themes addressed. I then offer a ‘roadmap’ of work in deep learning for computational bioacoustics, based on the survey and thematic analysis, and drawing in topics from the wider field of deep learning as well as broad topics in bioacoustics.

Survey Methodology

Deep learning is a recent and rapidly-moving field. Although deep learning has been applied to audio tasks (such as speech, music) for more than ten years, its application in wildlife bioacoustics is recent and immature. Key innovations in acoustic deep learning include Hershey et al. (2017) which represents the maturing of audio recognition based on convolutional neural networks (CNNs)—it introduced a dataset (AudioSet) and a NN architecture (VGGish) both now widely-used; and convolutional-recurrent neural network (CRNN) methods (Çakır et al., 2017). The organisers of the BirdCLEF data challenge announced “the arrival of deep learning” in 2016 (Goëau et al., 2016). Hence I chose to constrain keyword-based literature searches, using both Google Scholar and Web of Science, to papers published no earlier than 2016. The query used was: (bioacoust* OR ecoacoust* OR vocali* OR “animal calls” OR

“passive acoustic monitoring” OR “soundscape”) AND (“deep learning” OR “convolutional neural network” OR “recurrent neural network”) AND (animal OR bird* OR cetacean* OR insect* OR mammal*)

With Google Scholar, this search yielded 989 entries. Many were excluded due to being off-topic, or being duplicates, reviews, abstract-only, not available in English, or unavailable. Various preprints (non-peer reviewed) were encountered (in arXiv, biorXiv and more): I did not exclude all of these, but priority was given to peer-reviewed published work. With Web of Science, the same search query yielded 56 entries. After merging and deduplication, this yielded 162 articles. This sub-field is a rapidly-growing one: the set of selected articles grows from 5 in 2016 through to 64 in 2021. The bibliography file published along with this paper lists all these articles, plus other articles added for context while writing the review.

State of the Art and Recent Developments

I start with a standard recipe abstracted from the literature, and the taxonomic coverage of the literature, before reviewing those topics in bioacoustic DL that have received substantial attention and are approaching maturity. To avoid repetition, some of the more tentative or unresolved topics will be deferred to the later ‘roadmap’ section, even when discussed in existing literature.

The standard recipe for bioacoustic deep learning

Deep learning is flexible and can be applied to many different tasks, from classification/regression through to signal enhancement and even synthesis of new data. The ‘workhorse’ of DL is however classification, by which we mean assigning data items one or more ‘labels’ from a fixed list (e.g., a list of species, individuals or call types). This is the topic of many DL breakthroughs, and many other tasks have been addressed in part by using the power of DL classification—even image generation (Goodfellow et al., 2014). Classification is indeed the main use of DL seen in computational bioacoustics.

A typical ‘recipe’ for bioacoustic classification using deep learning, applicable to very many of the surveyed articles from recent years, is as follows. Some of the terminology may be unfamiliar, and I will expand upon it in later sections of the review: Use one of the well-known CNN architectures (ResNet, VGGish, Inception, MobileNet), perhaps pretrained from AudioSet. (These are conveniently available within the popular DL Python frameworks PyTorch, Keras, TensorFlow).

The input will be spectrogram data, typically divided into audio clips of fixed size such as 1 s or 10 s, which is done so that a ‘batch’ of spectrograms fits easily into GPU memory. The spectrograms may be standard (linear-frequency), or mel spectrograms, or log-frequency spectrograms. The “pixels” in the spectrogram are magnitudes: typically these are log-transformed before use, but might not be, or alternatively transformed by per-channel energy normalisation (PCEN). There is no strong consensus on the ‘best’ spectrogram format—it is likely a simple empirical choice based on the frequency bands of interest in your chosen task and their dynamic ranges.

The list of labels to be predicted could concern species, individuals, call types, or something else. It may be a binary (yes/no) classification task, which could be used for detecting the presence (occupancy) of some sound. In many cases a list of species is used: modern DL can scale to many hundreds of species. The system may be configured to predict a more detailed output such as a transcription of multiple sound events; I return to this later.

Use data augmentation to artificially make a small bioacoustic training dataset more diverse (noise mixing, time shifting, mixup).

Although a standard CNN is common, CRNNs are also relatively popular, adding a recurrent layer (LSTM or GRU) after the convolutional layers, which can be achieved by creating a new network from scratch or by adding a recurrent layer to an off-the-shelf network architecture.

Train your network using standard good practice in deep learning (for example: Adam optimiser, dropout, early stopping, and hyperparameter tuning) (Goodfellow, Bengio & Courville, 2016).

Following good practice, there should be separate data(sub)sets for training, validation (used for monitoring the progress of training and for selecting hyperparameters), and final testing/evaluation. It is especially beneficial if the testing set represents not just unseen data items but novel conditions, to better estimate the true generalisability of the system (Stowell et al., 2019b). However, it is still common for the training/validation/testing data to be sampled from the same pool of source data.

Performance is measured using standard metrics such as accuracy, precision, recall, F-score, and/or area under the curve (AUC or AUROC). Since bioacoustic datasets are usually “unbalanced”, having many more items of one category than another, it is common to account for this—for example by using macro-averaging, calculating performance for each class and then taking the average of those to give equal weight to each class (Mesaros, Heittola & Virtanen, 2016).

This standard recipe will work well for many bioacoustic classification tasks, including noisy outdoor sound scenes. (Heavy rain and wind remains a problem across all analysis methods, including DL). It can be implemented using a handful of well-known Python libraries: PyTorch/TensorFlow/Keras, librosa or another library for sound file processing, plus a data augmentation tool such as SpecAugment, audiomentations or kapre. The creation and data augmentation of spectrograms is specific to audio domains, but the CNN architecture and training is standard across DL for images, audio, video and more, which has the benefit of being able to inherit good practice from this wider field.

Data augmentation helps with small and also with unbalanced datasets, common in bioacoustics. The commonly-used augmentation methods (time-shifting, sound mixing, noise mixing) are “no regret” in that it is extremely unlikely these modifications will damage the semantic content of the audio. Other modifications such as time warping, frequency shifting, frequency warping, modify sounds in ways which could alter subtle cues such as those that might distinguish individual animals or call types from one another. Hence the appropriate choice of augmentation methods is audio-specific and even animal-sound-specific.

The standard recipe does though have its limits. The use of the mel frequency scale, AudioSet pretraining, and magnitude-based spectrograms (neglecting some details of phase or temporal fine structure) all bias the process towards aspects of audio that are easily perceptible to humans, and thus may overlook some details that are important for fine high-resolution discriminations or for matching animal perception (Morfi, Lachlan & Stowell, 2021a). All the common CNN architectures have small-sized convolutional filters, biasing them towards objects that are compact in the spectrogram, potentially an issue for broad-band sound events.

There are various works that pay close attention to the parameters of spectrogram generation, or argue for alternative representations such as wavelets. This engineering work can lead to improved performance in each chosen task, especially for difficult cases such as echolocation clicks. However, as has been seen in previous eras of audio analysis, these are unlikely to overturn standard use of spectrograms since the improvement rarely generalises across many tasks. Networks using raw waveforms as input may overcome many of these concerns in future, though they require larger training datasets; pretrained raw-waveform networks may be a useful tool to look forward to in the near term.

Taxonomic coverage

Species/taxa whose vocalisations have been analysed through DL include: Birds—the most commonly studied group, covered by at least 65 of the selected papers. Specific examples will be cited below, and some overviews can be found in the outcomes of data challenges and workshops (Stowell et al., 2019a, Joly et al., 2019).

Cetaceans and other marine mammals—another very large subfield, covered by 30 papers in the selected set. Again, data challenges and workshops are devoted to these taxa (Frazao, Padovese & Kirsebom, 2020).

Bats (Mac Aodha et al., 2018; Chen et al., 2020c, Fujimori et al., 2021; Kobayashi et al., 2021; Zhang et al., 2020; Zualkernan et al., 2020, 2021).

Terrestrial mammals (excluding bats): including primates (Bain et al., 2021; Dufourq et al., 2021; Romero-Mujalli et al., 2021; Oikarinen et al., 2019; Tzirakis et al., 2020), elephants (Bjorck et al., 2019), sheep (Wang et al., 2021), cows (Jung et al., 2021), koalas (Himawan et al., 2018).

A particular subset of work focusses on mouse and rat ultrasonic vocalisations (USVs). These have been of interest particularly in laboratory mice studies, hence a vigorous subset of literature based on rodent USVs primarily recorded in laboratory conditions (Coffey, Marx & Neumaier, 2019; Fonseca et al., 2021; Ivanenko et al., 2020; Premoli et al., 2021; Smith & Kristensen, 2017; Steinfath et al., 2021).

Anurans (Colonna et al., 2016; Dias, Ponti & Minghim, 2021; Hassan, Ramli & Jaafar, 2017; Islam & Valles, 2020; LeBien et al., 2020; Xie et al., 2021b, 2020, 2021c).

Insects (Hibino, Suzuki & Nishino, 2021; Khalighifar et al., 2021; Kiskin et al., 2021; Kiskin et al., 2020; Rigakis et al., 2021; Sinka et al., 2021; Steinfath et al., 2021).

Fish (Guyot et al., 2021; Ibrahim et al., 2018; Waddell, Rasmussen & Širović, 2021).

Many works cover more than one taxon, since DL enables multi-species recognition across a large number of categories and benefits from large and diverse data. Some works sidestep taxon considerations by focusing on the overall ecosystem or soundscape level (“ecoacoustic” approaches) (Sethi et al., 2020; Heath et al., 2021; Fairbrass et al., 2019).

The balance of emphasis across taxa has multiple drivers. Many of the above taxa are important for biodiversity and conservation monitoring (including birds, bats, insects), or for comparative linguistics and behaviour studies (songbirds, cetaceans, primates, rodents). For some taxa, acoustic communication is a rich and complex part of their behaviour, and their vocalisations have a high complexity which is amenable to signal analysis (Marler & Slabbekoorn, 2004). On the other hand, progress is undeniably driven in part by practical considerations, such as the relative ease of recording terrestrial and diurnal species. Aside from standard open science practices such as data sharing, progress in bird sound classification has been stimulated by large standardised datasets and automatic recognition challenges, notably the BirdCLEF challenge conducted annually since 2014 (Goëau et al., 2014; Joly et al., 2019). This dataset- and challenge-based progress follows a pattern of work seen in many applications of machine learning. Nonetheless, the allocation of research effort does not necessarily match up with the variety or importance of taxa—a topic I will return to.

Having summarised a standard recipe and the taxonomic coverage of the literature, I next review the themes that have received detailed attention in the literature on DL for computational bioacoustics.

Neural network architectures

The “architecture” of a neural network (NN) is the general layout of the nodes and their interconnections, often arranged in sequential layers of processing (Goodfellow, Bengio & Courville, 2016). Early work applying NNs to animal sound made use of the basic “multi-layer perceptron” (MLP) architecture (Koops, Van Balen & Wiering, 2015; Houégnigan et al., 2017; Hassan, Ramli & Jaafar, 2017; Mercado & Sturdy, 2017), with manually-designed summary features (such as syllable duration, peak frequency) as input. However, the MLP is superseded and dramatically outperformed by CNN and (to a lesser extent) recurrent neural network (RNN) architectures, both of which can take advantage of the sequential/grid structure in raw or lightly-preprocessed data, meaning that the input to the CNN/RNN can be time series or time-frequency spectrogram data (Goodfellow, Bengio & Courville, 2016). This change—removing the step in which the acoustic data is reduced to a small number of summary features in a manually-designed feature extraction process—keeps the input in a much higher dimensional format, allowing for much richer information to be presented. Neural networks are highly nonlinear and can make use of subtle variation in this “raw” data. CNNs and RNNs apply assumptions about the sequential/grid structure of the data, allowing efficient training. For example, CNN classifiers are by design invariant to time-shift of the input data. This embodies a reasonable assumption (most sound stimuli do not change category when moved slightly later/earlier in time), and results in a CNN having many fewer free parameters than the equivalent MLP, thus being easier to train.

One of the earliest works applying a CNN to bioacoustic data was to classify among 10 anuran species (Colonna et al., 2016). In the same year, 3 of the 6 teams in the 2016 BirdCLEF challenge submitted CNN systems taking spectrograms as input, including the highest-scoring team (Goëau et al., 2016). Reusing a high-performing CNN architecture from elsewhere is very popular now, but was possible even in 2016: one of the submitted systems re-used a 2012 CNN designed for images, called AlexNet. Soon after, Salamon et al. (2017a) and Knight et al. (2017) also found that a CNN outperformed the previous “shallow” paradigm of bioacoustic machine learning.

CNNs are now dominant: at least 83 of the surveyed articles made use of CNNs (sometimes in combination with other modules). Many articles empirically compare the performance of selected NN architectures for their tasks, and configuration options such as the number of CNN layers (Wang et al., 2021; Li et al., 2021; Zualkernan et al., 2020). Oikarinen et al. (2019) studied an interesting dual task of simultaneously inferring call type and caller ID from devices carried by pairs of marmoset monkeys, evaluating different types of output layer for this dual-task scenario.

While many of the surveyed articles used a self-designed CNN architecture, there is a strong move towards using, or at least evaluating, off-the-shelf CNN architectures (Lasseck, 2018; Zhong et al., 2020a; Guyot et al., 2021; Dias, Ponti & Minghim, 2021; Li et al., 2021; Kiskin et al., 2021; Bravo Sanchez et al., 2021; Gupta et al., 2021). These are typically CNNs that have been influential in DL more widely, and are now available conveniently in DL frameworks (Table 1). They can even be downloaded already pretrained on standard datasets, to be discussed further below. The choice of CNN architecture is rarely a decision that can be made from first principles, aside from general advice that the size/complexity of the CNN should generally scale with that of the task being attempted (Kaplan et al., 2020). Some of the popular recent architectures (notably ResNet and DenseNet) incorporate architectural modifications to make it feasible to train very deep networks; others (MobileNet, EfficientNet, Xception) are designed for efficiency, reducing the number of computations needed to achieve a given level of accuracy (Canziani, Paszke & Culurciello, 2016).

Table 1 Off-the-shelf CNN architectures used in the surveyed literature, and the number of articles using them.

This is indicative only, since not all articles clearly state whether an off-the-shelf model is used, some articles use modified/derived versions, and some use multiple architectures.

CNN architecture	Num articles	
ResNet (He et al., 2016)	23	
VGG (Simonyan & Zisserman, 2014) or VGGish (Hershey et al., 2017)	17	
DenseNet (Huang, Liu & Weinberger, 2016)	7	
AlexNet (Krizhevsky, Sutskever & Hinton, 2012)	5	
Inception (Szegedy et al., 2014)	4	
LeNet (LeCun et al., 1998)	3	
MobileNet (Sandler et al., 2018)	2	
EfficientNet (Tan & Le, 2019)	2	
Xception (Chollet, 2017)	2	
U-net (Ronneberger, Fischer & Brox, 2015)	2	
bulbul (Grill & Schlüter, 2017)	2	
Self-designed CNN	19	

The convolutional layers in a CNN layer typically correspond to non-linear filters with small “receptive fields” in the axes of the input data, enabling them to make use of local dependencies within spectrogram data. However, it is widely understood that sound scenes and vocalisations can be driven by dependencies over both short and very long timescales. This consideration about time series in general was the inspiration for the design of recurrent neural networks (RNNs), with the LSTM and GRU being popular embodiments (Hochreiter & Schmidhuber, 1997): these networks have the capacity to pass information forwards (and/or backwards) arbitrarily far in time while making inferences. Hence, RNNs have often been explored to process sound, including animal sound (Xian et al., 2016; Wang et al., 2021; Madhusudhana et al., 2021; Islam & Valles, 2020; Garcia et al., 2020; Ibrahim et al., 2018). An RNN alone is not often found to give strong performance. However, in around 2017 it was observed that adding an RNN layer after the convolutional layers of a CNN could give strong performance in multiple audio tasks, with an interpretation that the RNN layer(s) perform temporal integration of the information that has been preprocessed by the early layers (Cakir et al., 2017). This “CRNN” approach has since been applied variously in bioacoustics, often with good results (Himawan et al., 2018; Morfi & Stowell, 2018; Gupta et al., 2021; Xie et al., 2020; Tzirakis et al., 2020; Li et al., 2019). However, CRNNs can be more computationally intensive to train than CNNs, and the added benefit is not universally clear.

In 2016 an influential audio synthesis method entitled WaveNet showed that it was possible to model long temporal sequences using CNN layers with a special ‘dilated’ structure, enabling many hundreds of time steps to be used as context for prediction (van den Oord et al., 2016). This inspired a wave of work replacing recurrent layers with 1-D temporal convolutions, sometimes called temporal CNN (TCN or TCNN) (Bai, Kolter & Koltun, 2018). Note that whether applied to spectrograms or waveform data, these are 1-D (time only) convolutions, not the 2-D (time-frequency) convolutions more commonly used. TCNs can be faster to train than RNNs, with similar or superior results. TCNs have been used variously in bioacoustics since 2021, and this is likely to continue (Steinfath et al., 2021; Fujimori et al., 2021; Roch et al., 2021; Xie et al., 2021b; Gupta et al., 2021; Gillings & Scott, 2021; Bhatia, 2021). Gupta et al. (2021) compare CRNN against CNN+TCN, and also standard CNN architectures (ResNet, VGG), with CRNN the strongest method in their evaluation.

Innovations in NN architectures continue to be explored. Vesperini et al. (2018) apply capsule networks for bird detection. Gupta et al. (2021) apply Legendre memory units, a novel type of recurrent unit, in birdsong species classification. When we later review “object detection”, we will encounter some custom architectures for that task. In wider DL, especially text processing, it is popular to use a NN architectural modification referred to as “attention” (Chorowski et al., 2015). The structure of temporal sequences is highly variable, yet CNN and RNN architectures implicitly assume that the pattern of previous timesteps that are important predictors is fixed. Attention networks go beyond this by combining inputs in a weighted combination whose weights are determined on-the-fly. (Note that this is not in any strong sense a model of auditory attention as considered in cognitive science.) This approach was applied to spectrograms by Ren et al. (2018), and used for bird vocalisations by Morfi, Lachlan & Stowell (2021a). A recent trend in DL has been to use attention (as opposed to convolution or recurrence) as the fundamental building block of an NN architecture, known as “transformer” layers (Vaswani et al., 2017). Transformers are not yet widely explored in bioacoustic tasks, but given their strong performance in other domains we can expect their use to increase. The small number of recent studies shows encouraging results (Elliott et al., 2021; Wolters et al., 2021).

Many studies compare NN architectures empirically, usually from a manually-chosen set of options, perhaps with evaluation over many hyperparameter settings such as the number of layers. There are too many options to search them all exhaustively, and too little guidance on how to choose a network a priori. Brown, Montgomery & Garg (2021) propose one way to escape this problem: a system to automatically construct the workflow for a given task, including NN architecture selection.

Acoustic features: spectrograms, waveforms, and more

In the vast majority of studies surveyed, the magnitude spectrogram is used as input data. This is a representation in which the raw audio time series has been lightly processed to a 2D grid, whose values indicate the energy present at a particular time and frequency. Prior to the use of DL, the spectrogram would commonly be used as the source for subsequent feature extraction such as peak frequencies, sound event durations, and more. Using the spectrogram itself allows a DL system potentially to make use of diverse information in the spectrogram; it also means the input is a similar format to a digital image, thus taking advantage of many of the innovations and optimisations taking place in image DL.

Standard options in creating a spectrogram include the window length for the short-time Fourier transforms used (and thus the tradeoff of time- vs frequency-resolution), and the shape of the window function (Jones & Baraniuk, 1995). Mild benefits can be obtained by careful selection of these parameters, and have been argued for in DL (Heuer et al., 2019; Knight et al., 2020). A choice more often debated is whether to use a standard spectrogram with its linear frequency axis, or to use a (pseudo-)logarithmically-spaced frequency axis such as the mel spectrogram (Xie et al., 2019; Zualkernan et al., 2020) or constant-Q transform (CQT) (Himawan et al., 2018). The mel spectrogram uses the mel scale, originally intended as an approximation of human auditory selectivity, and thus may seem an odd choice for non-human data. Its use likely owes a lot to convenience, but also to the fact that pitch shifts of harmonic signals correspond to linear shifts on a logarithmic scale—potentially a good match for CNNs which are designed to detect linearly-shifted features reliably. Zualkernan et al. (2020) even found a mel spectrogram representation useful for bat signals, with of course a modification of the frequency range. The literature presents no consensus, with evaluations variously favouring the mel (Xie et al., 2019; Zualkernan et al., 2020), logarithmic (Himawan et al., 2018; Smith & Kristensen, 2017), or linear scale (Bergler et al., 2019b). There is likely no representation that will be consistently best across all tasks and taxa. Some studies take advantage of multiple spectrogram representations of the same waveform, by “stacking” a set of spectrograms into a multi-channel input (processed in the same fashion as the colour channels in an RGB image) (Thomas et al., 2019; Xie et al., 2021c). The channels are extremely redundant with one another; this stacking allows the NN flexibly to use information aggregated across these closely-related representations, and thus gain a small informational advantage.

ML practitioners must concern themselves with how their data are normalised and preprocessed before input to a NN. Standard practice is to transform input data to have zero mean and unit variance, and for spectrograms perhaps to apply light noise-reduction such as by median filtering. In practice, spectral magnitudes can have dramatically varying dynamic ranges, noise levels and event densities. Lostanlen et al. (2019a, 2019b) give theoretical and empirical arguments for the use of per-channel energy normalisation (PCEN), a simple adaptive normalisation algorithm. Indeed PCEN has been deployed by other recent works, and found to permit improved performance of deep bioacoustic event detectors (Allen et al., 2021; Morfi et al., 2021b).

As an aside, previous eras of acoustic analysis have made widespread use of mel-frequency cepstral coefficients (MFCCs), a way of compressing spectral information into a small number of standardised measurements. MFCCs have occasionally been used in bioacoustic DL (Colonna et al., 2016; Kojima et al., 2018; Jung et al., 2021). However, they are likely to be a poor match to CNN architectures since sounds are not usually considered shift-invariant along the MFCC coefficient axis. Deep learning evaluations typically find that MFCCs are outperformed by less-preprocessed representations such as the (closely-related) mel spectrogram (Zualkernan et al., 2020; Elliott et al., 2021).

Other types of time-frequency representation are explored by some authors as input to DL, such as wavelets (Smith & Kristensen, 2017; Kiskin et al., 2020) or traces from a sinusoidal pitch tracking algorithm (Jancovic & Köküer, 2019). These can be motivated by considerations of the target signal, such as chirplets as a match to the characteristics of whale sound (Glotin, Ricard & Balestriero, 2017).

However, the main alternative to spectrogram representations is in fact to use the raw waveform as input. This is now facilitated by NN architectures such as WaveNet and TCN mentioned above. DL based on raw waveforms is often found to require larger datasets for training than that based on spectrograms; one of the main attractions is to remove yet another of the manual preprocessing steps (the spectrogram transformation), allowing the DL system to extract information in the fashion needed. A range of recent studies use TCN architectures (also called 1-dimensional CNNs) applied to raw waveform input (Ibrahim et al., 2018; Li et al., 2019; Fujimori et al., 2021; Roch et al., 2021; Xie et al., 2021b). Ibrahim et al. (2018) compares an RNN against a TCN, both applied to waveforms for fish classification; Li et al. (2019) applies a TCN with a final recurrent layer to bird sound waveforms. Steinfath et al. (2021) offer either spectrogram or waveform input for their CNN segmentation method. Bhatia (2021) investigates bird sound synthesis using multiple methods including WaveNet. Transformer architectures can also be applied directly to waveform data (Elliott et al., 2021).

Some recent work has proposed trainable representations that are intermediate between raw waveform and spectrogram methods (Balestriero et al., 2018; Ravanelli & Bengio, 2018; Zeghidour et al., 2021). These essentially act as parametric filterbanks, whose filter parameters are optimised along with the other NN layer parameters. Balestriero et al. (2018) introduce their own trainable filterbank and achieve promising results for bird audio detection. Bravo Sanchez et al. (2021) applies a representation called SincNet, achieving competitive results on birdsong classification with a benefit of short training time. Zeghidour et al. (2021) apply SincNet but also introduce an alternative learnable audio front-end “LEAF”, which incorporates a trainable PCEN layer as well as a trainable filterbank, finding strong performance on a bird audio detection task.

To summarise this discussion: in many cases a spectrogram representation is appropriate for bioacoustic DL, often with (pseudo-)logarithmic frequency axis such as mel spectrogram or CQT spectrogram. PCEN appears often to be useful for spectrogram preprocessing. Methods using raw waveforms and adaptive front-ends are likely to gain increased prominence, especially if incorporated into some standard off-the-shelf NN architectures that are found to work well across bioacoustic tasks.

Classification, detection, clustering

The most common tasks considered in the literature, by far, are classification and detection. These tasks are fundamental building blocks of many workflows; they are also the tasks that are most comprehensively addressed by the current state of the art in deep learning.

The terms classification and detection are used in various ways, sometimes interchangeably. In this review I interpret ‘classification’ as in much of ML, the prediction of one or more categorical labels such as species or call type. Classification is very commonly investigated in bioacoustic DL. It is most widely used for species classification—typically within a taxon family, such as in the BirdCLEF challenge (Joly et al., 2021) (see above for other taxon examples). Other tasks studied are to classify among individual animals (Oikarinen et al., 2019; Ntalampiras & Potamitis, 2021), call types (Bergler et al., 2019a; Waddell, Rasmussen & Širović, 2021), sex and strain (within-species) (Ivanenko et al., 2020), or behavioural states (Wang et al., 2021; Jung et al., 2021). Some work broadens the focus beyond animal sound to classify more holistic soundscape categories such as biophony, geophony, anthropophony (Fairbrass et al., 2019; Mishachandar & Vairamuthu, 2021).

There are three different ways to define a ‘detection’ task that are common in the surveyed literature (Fig. 1):

Figure 1 Three common approaches to implementation of sound detection.

Adapted from Stowell et al. (2016b).

The first is detection as binary classification: for a given audio clip, return a binary yes/no decision about whether the signal of interest is detected within (Stowell et al., 2019a). This output would be described by a statistical ecologist as “occupancy” information (presence/absence). It is simple to implement since binary classification is a fundamental task in DL, and does not require data to be labelled in high-resolution detail. Perhaps for these reasons it is widely used in the surveyed literature (e.g. Mac Aodha et al., 2018; Prince et al., 2019; Kiskin et al., 2021; Bergler et al., 2019b; Himawan et al., 2018; Lostanlen et al., 2019b).

The second is detection as transcription, returning slightly more detail: the start and end times of sound events (Morfi et al., 2019; Morfi et al., 2021b). In the DCASE series of challenges and workshops (Detection and Classification of Acoustic Scenes and Events), the task of transcribing sound events, potentially for multiple classes in parallel, is termed sound event detection (SED), and in the present review I will use that terminology. It has typically been approached by training DL to label each small time step (e.g. a segment of 10 ms or 1 s) as positive or negative, and sequences of positives are afterwards merged into predicted event regions (Kong, Xu & Plumbley, 2017; Madhusudhana et al., 2021; Marchal, Fabianek & Aubry, 2021).

The third is the form common in image object detection, which consists of estimating multiple bounding boxes indicating object locations within an image. Transferred to spectrogram data, each bounding box would represent time and frequency bounds for an “object” (a sound event). This has not often been used in bioacoustics but may be increasing in interest (Venkatesh, Moffat & Miranda, 2021; Shrestha et al., 2021; Romero-Mujalli et al., 2021; Zsebök et al., 2019; Coffey, Marx & Neumaier, 2019).

For all three of these task settings, CNN-based networks are found to have strong performance, outperforming other ML techniques (Marchal, Fabianek & Aubry, 2021; Knight et al., 2017; Prince et al., 2019). Since the data format in each of the three task settings is different, the final (output) layers of a network take a slightly different form, as do the loss function used to optimise them (Mesaros et al., 2019). (Other settings are possible, for example pixel-wise segmentation of arbitrary spectral shapes (Narasimhan, Fern & Raich, 2017)).

The detection problem is important in the marine context, where surveys over very large scales of time and space are common, yielding very large data-processing demands, and the sounds to be detected may be very sparsely-occurring (Frazao, Padovese & Kirsebom, 2020). For this reason there are numerous works applying DL to detection of cetacean sounds underwater (Jiang et al., 2019; Bergler et al., 2019b; Best et al., 2020; Shiu et al., 2020; Zhong et al., 2020a; Ibrahim et al., 2021; Vickers et al., 2021b; Zhong et al., 2021; Roch et al., 2021; Vickers et al., 2021a; Allen et al., 2021; Madhusudhana et al., 2021).

In bioacoustics it is common to follow a two-step “detect then classify” workflow (Waddell, Rasmussen & Širović, 2021; LeBien et al., 2020; Schröter et al., 2019; Jiang et al., 2019; Koumura & Okanoya, 2016; Zhong et al., 2021; Padovese et al., 2021; Frazao, Padovese & Kirsebom, 2020; Garcia et al., 2020; Marchal, Fabianek & Aubry, 2021; Coffey, Marx & Neumaier, 2019). A notable benefit of the two-step approach is that for sparsely-occurring sounds, the detection stage can be tuned to reject the large number of ‘negative’ sound clips, with advantages for data storage/transmission, but also perhaps easing the process of training and applying the classifier, to make finer discriminations at the second step. Combined detection and classification is also feasible, and the SED and image object detection methods imported from neighbouring disciplines often include detection and classification within one NN architecture (Kong, Xu & Plumbley, 2017; Narasimhan, Fern & Raich, 2017; Shrestha et al., 2021; Venkatesh, Moffat & Miranda, 2021).

When no labels are available even to train a classifier, unsupervised learning methods can be applied such as clustering algorithms. The use of DL directly to drive clustering is not heavily studied. A typical approach could be to use an unsupervised algorithm such as an autoencoder (an algorithm trained to compress and then decode data); and then to apply a standard clustering algorithm to the autoencoder-transformed representation of the data, on the assumption that this representation will be well-behaved in terms of clustering similar items together (Coffey, Marx & Neumaier, 2019; Ozanich et al., 2021).

Signal processing using deep learning

Applications of DL have also been studied in computational bioacoustics, which do not come under the standard descriptions of classification, detection, or clustering. A theme common to the following less-studied tasks is that they relate variously to signal processing, manipulation or generation.

Denoising and source separation are preprocessing steps that have been used to improve the quality of a sound signal before analysis, useful in difficult signal-to-noise ratio (SNR) conditions (Xie, Colonna & Zhang, 2021a). For automatic analysis, it is worth noting that such preprocessing steps are not always necessary or desirable, since they may remove information from the signal, and DL recognition may often work well despite noise. Denoising and source separation typically use lightweight signal processing algorithms, especially when used as a front-end for automatic recognition (Xie, Colonna & Zhang, 2021a; Lin & Tsao, 2020). However, in many audio fields there is a move towards using CNN-based DL for signal enhancement and source separation (Manilow, Seetharman & Salamon, 2020). Commonly, this works on the spectrogram (rather than the raw audio). Instead of learning a function that maps the spectrogram onto a classification decision, denoising works by mapping the spectrogram onto a spectrogram as output, where the pixel magnitudes are altered for signal enhancement. DL methods for this are based on denoising autoencoders and/or more recently the u-net, which is a specialised CNN architecture for mapping back to the same domain (Jansson et al., 2017). In bioacoustics, some work has reported good performance of DL denoising as a preprocessing step for automatic recognition, both underwater (Vickers et al., 2021b; Yang et al., 2021) and for bird sound (Sinha & Rajan, 2018).

Privacy in bioacoustic analysis is not a mainstream issue. However, Europe’s General Data Protection Regulation (GDPR) drives some attention to this matter, which is well-motivated as acoustic monitoring devices are deployed in larger numbers and with increased sophistication (Le Cornu, Mitchell & Cooper, 2021). One strategy is to detect speech in bioacoustic recordings, in order to delete the respective recording clips, investigated for bee hive sound (Janetzky et al., 2021). Another is to approach the task as denoising or source separation, with speech the “noise” to remove. Cohen-Hadria et al. (2019) take this latter approach for urban sound monitoring, and to recreate the complete anonymised acoustic scene they go one step further by blurring the speech signal content and mixing it back into the soundscape. This is perhaps more than needed for most monitoring, but may be useful if the presence of speech is salient for downstream analysis, such as investigating human-animal interactions.

Data compression is another concern of relevance to deployed monitoring projects. If sound is to be compressed and transmitted back for some centralised analysis, there is a question about whether audio compression codecs will impact DL analysis. Heath et al. (2021) investigate this and concur with previous non-DL work that compression such as MP3 can have surprisingly small effect on analysis; they also obtain good performance using a CNN AudioSet embedding as a compressed ‘fingerprint’. Bjorck et al. (2019) use DL more directly to optimise a codec, producing a compressed representation of elephant sounds that (unlike the fingerprint) can be decoded to the audio clip.

Synthesis of animal sounds receives occasional attention, and could be useful among other things for playback experimental stimuli. Bhatia (2021) studies birdsong synthesis using modern DL methods, including a WaveNet and generative adversarial network (GAN) method.

Small data: data augmentation, pre-training, embeddings

The DL revolution has been powered in part by the availability of large labelled datasets. However, a widespread and persistent issue in bioacoustic projects is the lack of large labelled datasets: the species/calls may be rare or hard to capture, meaning not many audio examples are held; or the sound events may require a subject expert to annotate them with the correct labels (for training), and this expert time is often in short supply. Such constraints are felt for fine categorical distinctions such as those between conspecific individuals or call types, and also for large-scale monitoring in which the data volume far exceeds the person hours available. There are various strategies for effective work in such situations, including data mining and ecoacoustic methods; here I focus on techniques concerned with making DL feasible.

Data augmentation is a technique which artificially increases the size of a dataset (usually the training set) by taking the data samples and applying small irrelevant modifications to create additional data samples. For audio, this can include shifting the audio in time, adding low-amplitude noise, mixing audio files together (sometimes called ‘mixup’), or more complicated operations such as small warpings of the time or frequency axis in a spectrogram (Lasseck, 2018). The important consideration is that the modifications should not change the meaning (the label) of the data item. Importantly, in some animal vocalisations this may exclude frequency shifts. Data augmentation was in use even in 2016 at the “arrival” of DL (Goëau et al., 2016), and is now widespread, used in many of the papers surveyed. Various authors study the specific combinations of data augmentation, both for terrestrial and underwater sound (Lasseck, 2018; Li et al., 2021; Padovese et al., 2021). Data augmentation, using the basic set of augmentations mentioned above, should be a standard part of training most bioacoustic DL systems. Software packages are available to implement audio data augmentation directly (for example SpecAugment, kapre or audiomentions for Python). Beyond standard practice, data augmentation can even be used to estimate the impact of confounding factors in datasets (Stowell et al., 2019b).

A second widespread technique is pretraining: instead of training for some task by starting from a random initialisation of the NN, one starts from a NN that has previously been trained for some other, preferably related, task. The principle of “transfer learning” embodied here is that the two tasks will have some common aspects—such as the patterns of time-frequency correlations in a spectrogram, which at their most basic may have similar tendencies across many datasets—and that a NN can benefit from inheriting some of this knowledge gained from other tasks. This becomes particularly useful when large well-annotated datasets can be used for pretraining. Early work used pretraining from image datasets such as ImageNet, which gave substantial performance improvements even though images are quite different from spectrograms (Lasseck, 2018). Although ImageNet pretraining is still occasionally used (Disabato et al., 2021; Fonseca et al., 2021), many authors now pretrain using Google’s AudioSet (a diverse dataset of audio from YouTube videos (Hershey et al., 2017; Çoban et al., 2020; Kahl et al., 2021). A similar but more recent dataset is VGG-Sound (Chen et al., 2020a), used by Bain et al. (2021). Practically, off-the-shelf networks with these well-known datasets are widely available in standard toolkits. Although publicly-available bioacoustics-specific datasets (such as that from BirdCLEF) are now large, they are rarely explored as a source of pretraining—perhaps because they are not as diverse as AudioSet/VGG-Sound, or perhaps as a matter of convenience. Ntalampiras (2018) explored transfer learning from a music genre dataset. Contrary to the experiences of others, Morgan & Braasch (2021) report that pretraining was not of benefit in their task, perhaps because the dataset was large enough in itself (150 h annotated). Another alternative is to pretrain from simulated sound data, such as synthetic underwater clicks or chirps (Glotin, Ricard & Balestriero, 2017; Yang et al., 2021).

Closely related to pretraining is the popular and important concept of embeddings, and (related) metric learning. The common use of this can be stated simply: instead of using standard acoustic features as input, and training a NN directly to predict our labels of interest, we train a NN to convert the acoustic features into some partially-digested vector coordinates, such that this new representation is useful for classification or other tasks. The “embedding” is the space of these coordinates.

The simplest way to create an embedding is to take a pretrained network and remove the “head”, the final classification layers. The output from the “body” is a representation intermediate between the acoustic input and the highly-reduced semantic output from the head, and thus can often be a useful high-dimensional feature representation. This has been explored in bioacoustics and ecoacoustics using AudioSet embeddings, and found useful for diverse tasks (Sethi et al., 2021; Sethi et al., 2020; Çoban et al., 2020; Heath et al., 2021).

An alternative approach is to train an autoencoder directly to encode and decode items in a dataset, and then use the autoencoder’s learnt representation (from its encoder) as an embedding (Ozanich et al., 2021; Rowe et al., 2021). This approach can be applied even to unlabeled data, though it may not be clear how to ensure this encodes semantic information. It can be used as unsupervised analysis to be followed by clustering (Ozanich et al., 2021).

A third strategy for DL embedding is the use of so-called Siamese networks and triplet networks. These are not really a separate class of network architectures—typically a standard CNN is the core of the NN. The important change is the loss function: unlike most other tasks, training is not based on whether the network can correctly label a single item, but on the vector coordinates produced for a pair/triplet of items, and their distances from one another. In Siamese networks, training proceeds pairwise, with some pairs intended to be close together (e.g. same class) or far apart (e.g. different class). In triplet networks, training uses triplets with one selected as the ‘anchor’, one positive instance to be brought close, and one negative instance to be kept far away. In all cases, each of the items is projected through the NN independently, before the comparison is made. The product of such a procedure is this NN trained directly to produce an embedding in which location, or at least distance, carries semantic information. These and other embeddings can be used for downstream tasks by applying simple classification/clustering/regression algorithms to the learnt representation. A claimed benefit of Siamese/triplet networks is that they can train relatively well with small or unbalanced datasets, and this has been reported to be the case in terrestrial and underwater projects (Thakur et al., 2019; Nanni et al., 2020; Clementino & Colonna, 2020; Acconcjaioco & Ntalampiras, 2021; Zhong et al., 2021).

Other strategies to counter data scarcity have been investigated for bioacoustics: multi-task learning—another form of transfer learning, this involves training on multiple tasks simultaneously (Morfi & Stowell, 2018; Zeghidour et al., 2021; Cramer et al., 2020);

semi-supervised learning, which supplements labelled data with unlabelled data (Zhong et al., 2020b; Bergler et al., 2019a);

weakly-supervised learning, which allows for labelling that is imprecise or lacks detail (e.g. lacks start and end time of sound events) (Kong, Xu & Plumbley, 2017; Knight et al., 2017; Morfi & Stowell, 2018; LeBien et al., 2020);

self-supervised learning, which uses some aspect of the data itself as a substitute for supervised labelling. For example, Baevski et al. (2020) and Saeed, Grangier & Zeghidour (2021) present different self-supervised learning methods to pretrain a system, for use when large amounts of audio are available but no labels. In both of these works, the pretraining process optimises a NN to determine, for a given audio recording, which of a set of short audio segments genuinely comes from that recording. This contrastive learning task acts as a substitute for a truly semantic task, and performs well for speech and other audio. Since this is at heart an unsupervised learning approach, with no “guidance” on which aspects of the data are of interest, it remains to be seen how well it performs in bioacoustics where the key information may be only a small part of the overall energy of the signal;

few-shot learning, in which a system is trained across multiple similar tasks, in such a way that for a new unseen task (e.g. a new type of call to be detected) the system can perform well even with only one or very few examples of the new task (Morfi et al., 2021b; Acconcjaioco & Ntalampiras, 2021). A popular method for few-shot learning is to create embeddings using prototypical networks, which involve a customised loss function that aims to create an embedding having good “prototypes” (cluster centroids). Pons, Serrà & Serra (2019) determined this to outperform transfer learning for small-data scenarios, and it is the baseline considered in a recent few-shot learning bioacoustic challenge (Morfi et al., 2021b).

In general, these approaches are less commonly studied, and many authors in bioacoustics use off-the-shelf pretrained embeddings. However, many of the above techniques are useful to enable training despite dataset limitations; hence, they can themselves be used in creating embeddings, and could be part of future work on creating high-quality embeddings.

Generalisation and domain shift

Concern about whether the a DL system’s quality of performance will generalise to new data is a widespread concern, especially when small datasets are involved. A more specific concern is whether performance will generalise to new conditions in which attributes of the input data have changed: for example changes in the background soundscape, the sub-population of a species, the occurrence frequency of certain events, or the type of microphone used. All of these can change the overall distribution of basic acoustic attributes, so-called domain shift, which can have undesirable impacts on the outputs of data-driven inference (Morgan & Braasch, 2021).

It is increasingly common to evaluate DL systems, not only on a test set which is kept separate from the training data, but also on test set(s) which differ in some respects from the training data, such as location, SNR, or season (Shiu et al., 2020; Vickers et al., 2021b; Çoban et al., 2020; Allen et al., 2021; Khalighifar et al., 2021). This helps to avoid the risk of overestimating generalisation performance in practice.

Specific DL methods can be used explicitly to account for domain shift. Domain adaptation methods may automatically adapt the NN parameters (Adavanne et al., 2017; Best et al., 2020). Explicitly including contextual correlates as input to the NN is an alternative strategy for automatic adaptation (Lostanlen et al., 2019b, Roch et al., 2021). Where a small amount of human input about the new domain is possible, fine-tuning (limited retraining) or active learning (interactive feedback on predictions) have been explored (Çoban et al., 2020; Allen et al., 2021; Ryazanov et al., 2021). Stowell et al. (2019b) designed a public “bird audio detection” challenge specifically to stimulate the development cross-condition (cross-dataset) generalisable methods. In that challenge, however, the leading submissions did not employ explicit domain adaptation, instead relying on the implicit generality of transfer learning (pretraining) from general-purpose datasets, as well as data augmentation to simulate diverse conditions during training.

Open-set and novelty

One problem with the standard recipe (and in fact many ML methods) is that by default, recognition is limited to a pre-specified and fixed set of labels. When recording in the wild, it is surely possible to encounter species or individuals not accounted for in the training set, which should be identified. This is common e.g. for individual ID (Ptacek et al., 2016). Detecting new sound types beyond the known set of target classes is referred to as open set recognition, perhaps related to the more general topic of novelty detection which aims to detect any novel occurrence in data. Cramer et al. (2020) argue that hierarchical classification is useful for this, in that a sound may be strongly classified to a higher-level taxon even when the lower-level class is novel. Ntalampiras & Potamitis (2021) apply novelty detection based on a CNN autoencoder (an algorithm trained to compress and then recreate data). Since the method is trained to reconstruct the training examples with low error, the authors use the assumption that novel sounds will be reconstructed with high error, and thus use this as a trigger for detecting novelty.

More broadly, the aforementioned topic of embeddings offers a useful route to handling open-set classification. A good embedding should provide a somewhat semantic representation even of new data, such that even novel classes will cluster well in the space (standard clustering algorithms such as k-nearest neighbours can be applied). This is advocated by Thakur et al. (2019), using triplet learning, and later Acconcjaioco & Ntalampiras (2021) using Siamese learning. Novelty and open-set issues are likely to be an ongoing concern, in practice if not in theory, though the increasing popularity of general-purpose embeddings indeed offers part of the solution.

Context and auxiliary information

Deep learning implementations almost universally operate on segments of audio or spectrogram (e.g. 1 or 10 s per datum) rather than a continuous data stream. This is true even for RNNs which in theory can have unbounded time horizons. Yet it is clear from basic considerations that animal vocalisations, and their accurate recognition, may depend strongly on contextual factors originating outside a short window of temporal attention, whether this be prior soundscape activity or correlates such as date/time, location or weather.

Lostanlen et al. (2019b) add a “context-adaptive neural network” layer to their CNN, whose weights are dynamically adapted at prediction time by an auxiliary network taking long-term summary statistics of spectrotemporal features as input. Similarly, Roch et al. (2021) input acoustic context to their CNN based on estimates of the local signal-to-noise ratio (SNR). Madhusudhana et al. (2021) apply a CNN (DenseNet) to acoustic data, and then postprocess the predictions of that system using an RNN, to incorporate longer-term temporal context into the final output. Note that this CNN and RNN is not an integrated CRNN but two separate stages, with the consequence that the RNN can be applied over differing (longer) timescales than the CNN.

Animal taxonomy is another form of contextual information which may help to inform or constrain inferences. Although taxonomy is rarely the strongest determinant of vocal repertoire, it may offer a partial guide. Hierarchical classification is used in many fields, including bioacoustics; Cramer et al. (2020) propose a method that explicitly encodes taxonomic relationships between classes into the training of a CNN, evaluated using bird calls and song. Nolasco & Stowell (2022) propose a different method, and evaluate across a broader hierarchy, covering multiple taxa at the top level and individual animal identity at the lowest level.

Perception

The work so far discussed uses DL as a practical tool. Deep learning methods are loosely inspired by ideas from natural perception and cognition (LeCun, Bengio & Hinton, 2015), but there is no strong assumption that bioacoustic DL implements the same processes as natural hearing. Further, since current DL models are hard to interpret, it would be hard to validate whether or not that assumption held.

Even so, a small stream of research aims to use deep learning to model animal acoustic perception. DL can model highly non-linear phenomena, so perhaps could replicate many of the subtleties of natural hearing, which simpler signal processing models do not. Such models could then be studied or used as a proxy for animal judgment. Morfi, Lachlan & Stowell (2021a) use triplet loss to train a CNN to produce the same decisions as birds in a two-alternative forced-choice experiment. Simon et al. (2021) train a CNN from sets of bat echolocation call reflections, to classify flowers as bat-pollinated or otherwise—a simplified version of an object recognition task that a nectarivorous/frugivorous bat presumably solves. Francl & McDermott (2020) study sound localisation, finding that a DL trained to localise sounds in a (virtual) reverberant environment exhibits some phenomena known from human acoustic perception.

On-device deep learning

Multiple studies focus on how to run bioacoustic DL on a small hardware device, for affordable/flexible monitoring in the field. Many projects do not need DL running in real-time on device: they can record audio to storage or transmit it to a base station, for later processing (Roe et al., 2021; Heath et al., 2021). However, implementing DL on-device allows for live readouts and rapid responses (Mac Aodha et al., 2018), potential savings in power or data transmission costs, and enables some patterns of deployment that might not otherwise be possible. One benefit of wide interest might be to perform a first step of detection/filtering and discard many hours of uninformative audio, to extend deployment durations before storage is full and reduce transmission bandwidths: this is traditionally performed with simple energy detection, but could be enhanced by lightweight ML algorithms, perhaps similar to “keyword spotting” in domestic devices (Zhang et al., 2017).

The Raspberry Pi is a popular small Linux device, and although low-power it can have much of the functionality of a desktop computer, such as running Python or R scripts; thus the Raspberry Pi has been used for acoustic monitoring and other deployments (Jolles, 2021). Similar devices are NVIDIA Jetson Nano and Google Coral (the latter carries a TPU unit on board intended for DL processing). Zualkernan et al. (2021) evaluate these three for running a bat detection algorithm on-device.

Even more constrained devices offer lower power consumption (important for remote deployment powered by battery or solar power), lower ecological footprint, and smaller form factor; often based on the ARM Cortex-M family of processors. The AudioMoth is a popular example (Hill et al., 2018). It is too limited to run many DL algorithms; however Prince et al. (2019) were able to implement a CNN (depthwise-separable to reduce the complexity), applied to mel frequency features, and report that it outperformed a hidden Markov model detector on-device, although “at the cost of both size and speed”: it was not efficient enough to run in real-time on AudioMoth. Programming frameworks help to make such low-level implementations possible: Disabato et al. (2021) use ARM CMSIS (Common Microcontroller Software Interface Standard—a software interface) to implement a bird detector, and Zualkernan et al. (2021) use TensorFlow Lite to implement a bat species classifier. As in the more general case, off-the-shelf NN architectures can be useful, including MobileNet and SqueezeNet which are designed to be small/efficient (Vidaña-Vila et al., 2020). However, all three of the bioacoustic studies just mentioned, while inspired by these, implemented their own CNN designs and feature modifications to shrink the footprint even further.

Small-footprint device implementations offer the prospect of DL with reduced demands for power, bandwidth and storage. However, Lostanlen et al. (2021b) argue that energy efficiency is not enough, and that fundamental resource requirements such as the rare minerals required for batteries are a constraint on wider use of computational bioacoustic monitoring. They propose batteryless acoustic sensing, using novel devices capable of intermittent computing whenever power becomes available. It remains to be seen whether this intriguing proposal can be brought together with the analytical power of DL.

Workflows and other practicalities

As DL comes into increased use in practice, questions shift from the proof-of-concept to the integration into broader workflows (e.g. of biodiversity monitoring), and other practicalities. Many of the issues discussed above arise from such considerations. Various authors offer recommendations and advice for ecologists using DL (Knight et al., 2017; Rumelt, Basto & Roncal, 2021; Maegawa et al., 2021). Others investigate integration of a CNN detector/classifier into an overall workflow including data acquisition, selection and labeling (LeBien et al., 2020; Morgan & Braasch, 2021; Ruff et al., 2021). Brown, Montgomery & Garg (2021) go further and investigate the automation of designing the overall workflow, arguing that “[t]here is merit to searching for workflows rather than blindly using workflows from literature. In almost all cases, workflows selected by [their proposed] search algorithms (even random search, given enough iterations) outperformed those based on existing literature.”

One aspect of workflow is the user interface (UI) through which an algorithm is configured and applied, and its outputs explored. Many DL researchers provide their algorithms as Python scripts or suchlike, a format which is accessible by some but not by all potential users. Various authors provide graphical user interfaces (GUIs) for the algorithms they publish, and to varying extents study efficient graphical interaction (Jiang et al., 2019; Coffey, Marx & Neumaier, 2019; Steinfath et al., 2021; Ruff et al., 2021).

A Roadmap for Bioacoustic Deep Learning

I next turn to the selection of topics that are unresolved and/or worthy of further development: recommended areas of focus in the medium-term for research in deep learning applied within computational bioacoustics. The gaps are identified and confirmed through the literature survey, although there will always be a degree of subjectivity in the thematic synthesis.

Let us begin with some principles. Firstly, AI does not replace expertise, even though this may be implied by the standard recipe and general approach (i.e. using supervised learning to reproduce expert labels). Instead, through DL we train sophisticated but imperfect agents, with differing sets of knowledge. For example, a bird classifier derived from an AudioSet embedding may have one type of expertise, while a raw waveform system trained from scratch has a different expertise. As the use of these systems becomes even more standardised, they take on the role of expert peers, with whom we consult and debate. The move to active learning, which deserves more attention, cements this role by allowing DL agents to learn from criticism of their decisions. Hence, DL does not displace the role of experts, nor even of crowdsourcing; future work in the field will integrate the benefits of all three (Kitzes & Schricker, 2019). Secondly, open science is a vital component of progress. We have seen that the open publication of datasets, NN architectures, pretrained weights, and other source code has been crucial in the development of bioacoustic DL. There is a move toward open sharing of data, but in bioacoustics this is incomplete (Baker & Vincent, 2019). Sharing audio and metadata, and the standardisation of metadata, will help us to move far beyond the limitations of single datasets.

Maturing topics? Architectures and features

Let us also briefly revise core topics within bioacoustic DL that are frequently discussed, but can be considered to be maturing, and thus not of high urgency.

The vast majority of the surveyed work uses spectrograms or mel spectrograms as the input data representation. Although some authors raise the question of whether a species-customised spectrogram should be more appropriate than the human-derived mel spectrogram, for many tasks such alterations are unlikely to make a strong difference: as long as the spectrogram represents sufficient detail, and a DL algorithm can reasonably be trained, the empirical performance is likely to be similar across many different spectrogram representations. Preprocessing such as noise reduction and PCEN is often found to be useful and will continue to be applied. Methods based on raw waveforms, or adaptive front-ends such as SincNet or LEAF, are certainly of interest, and further exploration of these in bioacoustics is anticipated. They may be particularly useful for tasks requiring fine-grained distinctions.

Commonly-used acoustic “features” in future are likely to include off-the-shelf deep embeddings, even more commonly than now. Whether the input to those features is waveform or spectrogram will be irrelevant to users. AudioSet and VGG-Sound are the most commonly-used datasets for such pretraining; note however that these cannot cover all bioacoustic needs—for example ultrasound—and so it seems likely that bioacoustics-specific embeddings will be useful in at least some niches.

CNNs have become dominant in many applications of DL, and this applies to bioacoustic DL too. The more recent use of one-dimensional temporal convolutions (TCNs) is likely to continue, because of their simplicity and relative efficiency. However, looking forward it is not in fact clear whether CNNs will retain their singular dominance. In NLP and other domains, NN architectures based on “attention” (transformers/perceivers, discussed above) have displaced CNN as a basic architecture. CNNs fit well with waveform and spectrogram data, and thus are likely to continue to contribute to NN architectures for sound, perhaps combined with transformer layers (cf. Baevski et al. (2020)). For example, Wolters et al. (2021) propose to address sound event detection by using a CNN together with a perceiver network: their results imply that a perceiver is a good way to process variable-length spectrogram data into per-event summary representations.

A similar lesson applies to RNNs, except that RNNs have a more varied history of popularity. Recent CRNNs make good use of recurrent layers; but TCNs seem to threaten to displace them. I suggest that although RNNs embody a very general idea about sequential data, they are a special case of more general computation with memory. Transformers and other attention NNs show a different approach to allowing a computation to refer back to previous time steps flexibly. (See also the Legendre memory unit explored by Gupta et al. (2021).) All are special cases, and future work in DL may move more towards the general goal of differentiable neural computing (Graves, Wayne & Danihelka, 2014). The fluctuating popularity of recurrence and attention depends on their convenience and reusability as efficient modules in a DL architecture, and the future of DL-with-memory is likely to undergo many changes. Computational bioacoustics will continue to use these and integrate short- and long-term memory with other contextual data.

Learning without large datasets

Bioacoustics in general will benefit from the increasing open availability of data. However, this does not dissipate the oft-studied issue of small data: project-specific recognition tasks will continue to arise, including high-resolution discrimination tasks, and tasks for which transfer learning is inappropriate (e.g. due to the risk of bias) (Morfi, Lachlan & Stowell, 2021a). Many approaches to dealing with small datasets have been surveyed in the preceding text; important for future work is for these approaches to be integrated together, and for their advantages and disadvantages to be clarified. As shown in data challenges such as BirdCLEF and DCASE, pre-training, embeddings, multi-task learning and data augmentation all offer low-risk methods for improved generalisation.

Few-shot learning is a recent topic of interest; it is not clear whether it will continue long-term to be a separate “task” or will integrate with wider approaches, but it reflects a common need in bioacoustic practice. Active learning (AL) is also a paradigm of recent interest, and of high importance. It moves beyond the basic non-interactive model of most machine learning, in which a fixed training set is the only information available to a classifier. In AL, there is a human-machine interaction of multiple iterations, in which (some) predictions from a system are shown to a user for feedback, and the user’s feedback about correct and mistaken identifications is fed into the next round of optimisation of the algorithm. I identify AL as high importance because it offers a principled way to make the most efficient use of a person’s time in labelling or otherwise interacting with a system (Qian et al., 2017). It can be a highly effective way to deal with large datasets, including domain shift and other issues. It has been used in bioacoustic DL (Steinfath et al., 2021; Allen et al., 2021) but is under-explored, in part because its interactive nature makes it slightly more complex to design an AL evaluation. It may be that future work will use something akin to few-shot learning as the first step in an AL process.

A very different approach to reduce the dependence on large data is to create entirely simulated datasets that can be used for training. This is referred to as sim2real in DL, and its usefulness depends on whether it is feasible to create good simulations of the phenomena to be analysed. It goes beyond data augmentation in generating new data points rather than modifying existing ones. It may thus be able to generate higher diversity of training data, at a cost of lower realism. One notable advantage of sim2real is that any confounds or biases in the training data can be directly controlled. Simulated datasets have been explored in training DL detectors of marine sounds, perhaps because this class of signals can be modelled using chirp/impulse/sinusoidal synthesis (Glotin, Ricard & Balestriero, 2017; Yang et al., 2021; Li et al., 2020). Simulation is also especially relevant for spatial sound scenes, since the spatial details of natural sound scenes are hard to annotate (Gao et al., 2020; Simon et al., 2021). Simulation, often involving composing soundscapes from a library of sound clips, has been found useful in urban and domestic sound analysis (Salamon et al., 2017b; Turpault et al., 2021). Such results imply that wider use in bioacoustic DL may be productive, even when simulation of the sound types in question is not perfect.

Equal representation

Deep learning systems are well-known to be powerful but with two important weaknesses: (1) in most cases they must be treated as ‘black boxes’ whose detailed behaviour in response to new data is an empirical question; (2) they can carry a high risk of making biased decisions, which usually occurs because they faithfully reproduce biases in the training data (Koenecke et al., 2020). Our concern here is to create DL systems that can be a reliable guide to animal vocalisations, especially if used as a basis for conservation interventions. Hence we should ensure that the tools we create lead to an equal representation in terms of their sensitivity, error rates, etc. (Hardt et al., 2016).

Baker & Vincent (2019) point out that research output in bioacoustics is strongly biased: its taxonomic balance is unrepresentative of the audible animal kingdom, whether considered in terms of species diversity, biomass, or conservation importance. The same is true in the sub-field of DL applied to bioacoustics, both for datasets and research papers (see taxa listed above). Baker advocates for further attention to insect sound, and insects are recognised more broadly as under-studied (Montgomery et al., 2020); Linke et al. (2018) make a related case for freshwater species.

Equal representation (taxonomic, geographic, etc.) can be inspected in a dataset, and we should make further efforts to join forces and create more diverse open datasets, covering urban and remote locations, rich and poor countries. Baker & Vincent (2019) argue that the general field of bioacoustic research suffers from a lack of data deposition, with only 21% of studied papers publishing acoustic recordings for others to use. Addressing this gap in open science practice may in fact be our most accessible route to better coverage in acoustic data.

Equal representation should also be evaluated in feature representations such as widely-used embeddings. The representational capacities of an embedding derive from the dataset, the NN architecture and the training regime, and any of these factors could introduce biases that represent some acoustic behaviours better than others.

Beyond equal representation, it may of course remain important to develop targeted methods, such as those targeted at rare species (Znidersic et al., 2020; Wood et al., 2021). Since rare occurrences are intrinsically difficult to create large datasets for, this is worthy of further study. This review lists many methods that may help when rare species are of interest, but the best use of them is not yet resolved. To give examples beyond the bioacoustic literature: Beery et al. (2020) explore the use of synthetic examples for rarely-observed categories (in camera trap images); and Baumann et al. (2020) consider frameworks for evaluating rare sound event detection.

Interfaces and visualisation

Many bioacoustic DL projects end with their outputs as custom Python scripts: this is good practice in computer science/DL, for reproducibility, but not immediately accessible to a broad community of zoologists/conservationists. User interfaces (UIs) are a non-trivial component in bridging this gap. Since the potential users of DL may wish to use it via R, Python, desktop apps, smartphone apps, or websites, there remains no clear consensus on what kinds of UI will be most appropriate for bioacoustic DL, besides the general wish to integrate with existing audio editing/annotation tools. It seems likely that in future many algorithms will be available as installable packages or web APIs, and accessed variously through R/Python/desktop/etc. as preferred. Some existing work creates and even evaluates interfaces (discussed above, Workflow section), but more work on this is merited, including (a) research on efficient human-computer interaction for bioacoustics, and (b) visualisation tools making use of large-scale DL processing (cf. Kholghi et al., 2018; Znidersic et al., 2020; Phillips, Towsey & Roe, 2018).

One domain in which user interaction is particularly important is active learning (AL), since it involves an iterative human-computer interaction. The machine learning components in AL can be developed without UI work, but interaction with sound data has idiosyncratic characteristics (temporal regions, spectrograms, simultaneously-occurring sounds) which suggest that productive bioacoustic AL will involve UI designs that specifically enhance this interaction.

Beyond human-computer interaction is animal-computer interaction, for example using robotic animal agents in behavioural studies (Simon et al., 2019; Slonina et al., 2021). These studies offer the prospect of new insights about animal behaviour, and they might use DL in future to provide sophisticated vocal interaction.

The most common formulation of DL tasks, via fixed sets of training data and evaluation data, become less relevant when considering active learning and other interactive situations. There will need to be further consideration of the format, for example of data-driven challenges, and potentially DL techniques such as reinforcement learning (not reviewed here since not used in the current literature) (Teşileanu, Ölveczky & Balasubramanian, 2017).

Under-explored machine learning tasks

The following tasks are known in the literature, but according to the present survey are not yet mature, and also worthy of further work because of their importance or generality.

Individual ID

Automatically recognising discriminating between individual animals has been addressed by many studies in bioacoustics, whether for understanding animal communication or for monitoring/censusing animals (Ptacek et al., 2016; Vignal, Mathevon & Mottin, 2008; Searby, Jouventin & Aubin, 2004; Linhart et al., 2019; Adi, Johnson & Osiejuk, 2010; Fox, 2008; Beecher, 1989). Acoustic recognition of individuals can be a non-invasive replacement for survey techniques involving physical capture of individuals; it thus holds potential for improved monitoring with lower disturbance of wild populations. Thus far DL has only rarely been applied to individual ID in acoustic surveying, though this will surely change (Ntalampiras & Potamitis, 2021; Nolasco & Stowell, 2022). For some species, inter-individual distinctions may be as simple as a difference in fundamental frequency, and thus addressable using a simple algorithm; however, in general the attraction for DL here comes from its capacity to distinguish highly nonlinear patterns and its potential to generalise well to out-of-sample data. Within-species acoustic differences between individuals are typically fine-scale differences, requiring finer distinctions than species distinctions. This makes the task harder than species classification.

I suggest that these characteristics make the task of general-purpose automatic discrimination of individual animals, a useful focus for DL development. A DL system that can address this task usefully is one that can make use of diverse fine acoustic distinctions. Its inferences will be of use in ethology as well as in biodiversity monitoring. Cross-species approaches and multi-task learning can help to bridge bioacoustic considerations across the various taxon groups commonly studied (Nolasco & Stowell, 2022). A complete approach to individual recognition would also handle the open-set issue well, since novel individuals may often be encountered in the wild. There are not many bioacoustic datasets labelled with individual ID, and increased open data sharing can help with this.

Sound event detection and object detection

For many reasons it can be useful to create a detailed “transcript” of the sound events within a recording, going beyond basic classification. As with individual ID, this more detailed analysis can feed into both ethological and biodiversity analyses; its development goes hand-in-hand with higher-resolution bioacoustic DL.

As described in the earlier discussion of detection, there are at least two main approaches to this in existing literature. One version of SED (Fig. 1B) follows the same model as automatic music transcription or speaker diarisation in other domains, and uses similar DL architectures to solve the problem (Mesaros et al., 2021; Morfi & Stowell, 2018; Morfi et al., 2021b). An alternative approach inherits directly from object detection based on bounding boxes in images (Fig. 1C). This fits well when data annotations are given as time-frequency bounding boxes drawn on spectrograms. Solutions typically adapt well-known image object detector architectures such as YOLO and Faster R-CNN, which are quite different from the architectures used in other tasks (Venkatesh, Moffat & Miranda, 2021; Shrestha et al., 2021; Zsebök et al., 2019; Coffey, Marx & Neumaier, 2019). These two approaches each have their advantages. For example, frequency ranges in sound events can sometimes be useful information, but can sometimes be ill-defined/unnecessary, and not present in many datasets. Future work in bioacoustic DL should take the best of each paradigm, perhaps with a unified approach that can be applied whether or not frequency bounds are included in the data about a sound event.

Spatial acoustics

On a fine scale, the spatial arrangement of sound sources in a scene can be highly informative, for example in attributing calls to individuals and/or counting individuals correctly. It can also be important for behavioural and evolutionary ecological analysis (Jain & Balakrishnan, 2011). Spatial location can be handled using multi-microphone arrays, including stereo or ambisonic microphones. It is often analysed in terms of the direction-of-arrival (DoA) and/or range (distance) relative to the sensor. Taken together, the DoA and the range imply the Cartesian location; but either of them can be useful on its own.

The standard approach to spatial analysis uses signal processing algorithms, even when the data are later to be classified using machine learning (Kojima et al., 2018). However, this may change. For example, Houégnigan et al. (2017) train an MLP and Van Komen et al. (2020) a CNN, to estimate the range (distance) of underwater synthetic sound events. Yip et al. (2019) perform a similar task using terrestrial recordings: using calibrated microphone recordings of two bird species, they obtain useful estimates of distance by deep learning regression from the sound level measurements. In other domains of acoustics e.g. speech and urban sound, there is already a strong move to supplant signal processing with DL for spatial tasks (Hammer et al., 2021; Adavanne et al., 2018; Shimada et al., 2021). Important to note is that these works usually deal with indoor sound. Indoor and outdoor environments have very different acoustic propagation effects, meaning that the generalisation to outdoor sound may not be trivial (Traer & McDermott, 2016).

Spatial inference can also be combined with SED (e.g. in the DCASE challenge “sound event localisation and detection” task or SELD), combining the two-step process (e.g. Kojima et al., 2018) into a single joint estimation task (Shimada et al., 2021).

It is clear that many bioacoustic datasets and research questions will continue to be addressed in a spatially-agnostic fashion. Although some spatial attributes such as distance can be estimated from single-channel recordings (as above), multi-channel audio is usually required for robust spatial inference. Spatial acoustic considerations are quite different in terrestrial and marine sound, and more commonly considered in the latter. However, the development of DL tasks such as distance estimation and SELD (in parallel to SED) could benefit bioacoustics generally, with local spatial information used more widely in analysis.

The discussion thus far does not address the broader geographic-scale distribution of populations, which statistical ecologists may estimate from observations. Although machine observations will increasingly feed into such work, the use of DL in statistical ecology is outside the scope of this review (but cf. Kitzes & Schricker, 2019).

Useful integration of outputs

As DL becomes increasingly used in practice, there will inevitably be further work on integrating it into practical workflows, discussed earlier. However, there are some gaps to be bridged, worthy of specific attention.

An important issue is the calibration of the outputs of automatic inference. Kitzes & Schricker (2019) state the problem: “We wish to specifically highlight one subtler challenge, however, which we believe is substantially hindering progress: the need for better approaches for dealing with uncertainty in these indirect observations. […] First, machine learning classifiers must be specifically designed to return probabilistic, not binary, estimates of species occurrence in an image or recording. Second, statistical models must be designed to take this probabilistic classifier output as input data, instead of the more usual binary presence–absence data. The standard statistical models that are widely used in ecology and conservation, including generalized linear mixed models, generalized additive models and generalized estimating equations, are not designed for this type of input.” (Kitzes & Schricker, 2019)

In fact, although many ML algorithms do output strict binary decisions, DL classifiers/detectors do not: they output numerical values between zero and one, which we can interpret as probabilities, or convert into binary decisions by thresholding. However, the authors’ first point does not disappear since the outputs from DL systems are not always well-calibrated probabilities: they may under- or over-confident, depending on subtleties of how they have been trained (such as regularisation) (Niculescu-Mizil & Caruana, 2005). This does not present an issue when evaluating DL by standard metrics, but becomes clear when combining many automatic detections to form abundance estimates. DL outputs, interpreted as probabilities, may be under- or over-confident, or biased in favour of some categories and against others. Measuring (mis)calibration is the first step, and postprocessing the outputs can help (Niculescu-Mizil & Caruana, 2005). Evaluating systematic biases is also important: DL can be expected to exhibit higher sensitivity towards sounds well-represented in its training data, and this has been seen in practice (Lostanlen et al., 2018). Birdsong species classifiers are strongest for single-species recordings, and even with current DL they show reduced performance in denser soundscape recordings—an important concern given that much birdsong is heard in dense dawn choruses (Joly et al., 2019). Evaluating and improving upon these biases is vital.

The spatial reliability of detection is one particular facet of this. For manual surveys, there is well-developed statistical methodology to measure how detection probability relates to the distance to the observer, and how this might vary with species and habitat type (Johnston et al., 2014). The same must be applied to automatic detectors. We have an advantage of reproducibility: we can assume that distance curves and calibration curves for a given DL algorithm, analysing audio from a given device model, will be largely consistent. Thus such measurements applied to a widely-used DL algorithm and recording device would be widely useful. Some work does evaluate the performance of bioacoustic DL systems and how they degrade over distance (Maegawa et al., 2021; Lostanlen et al., 2021a). This can be developed further, in both simulated and real acoustic environments.

Under the model of detection as binary classification, our observations are “occupancy” (presence/absence) measurements. These can be used to estimate population distributions, but are less informative than observed abundances of animals. Under the more detailed models of detection, we can recover individual calls/song bouts and then count them, though of course these do not directly reflect the number of animals unless we can use calling-rate information collected separately (Stevenson et al., 2015). Routes toward bridging this gap using DL include applying “language models” of vocal sequences and interactions; the use of spatial information to segregate calls per individual; and direct inference of animal abundance, skipping the intermediate step of call detections. Counting and density estimation using DL has been explored for image data (e.g. Arteta, Lempitsky & Zisserman, 2016), a kind of highly nonlinear regression task. Early work explores this for audio, using a CNN to predict the numbers of targeted bird/anuran species in a sound clip (Dias, Ponti & Minghim, 2021). Sethi et al. (2021) suggest that regression directly from deep acoustic embedding features to species relative density can work well, especially for common species with strong temporal occurrence patterns.

As DL tools become integrated into various workflows, the issue of standardised data exchange becomes salient. Standards bodies such as Biodiversity Information Standards (“TDWG”) provide guidance on formats for biodiversity data exchange, including those for machine observations (https://www.tdwg.org/). These standards are useful, but may require further development: for example, the probabilistic rather than binary output referenced by Kitzes & Schricker (2019) needs to be usefully represented. The attribution of observations to a specific algorithm (trained using specific datasets…) requires a refinement of the more conventional attribution metadata schemes used for named persons. Such attribution can perhaps already be represented by standards such as the W3C Provenance Ontology, though such usage is not widespread (https://www.w3.org/TR/prov-o/). An explicit machine-readable representation of such provenance relationships would greatly simplify tasks such as merging of multiple machine-observation datasets, or investigating the causal effects of algorithmic and training-data choices on deployed systems.

Taken together, these integration-related technical topics are important for closing the loop between bioacoustic monitoring, data repositories, policy and interventions. They are thus salient for bringing bioacoustic DL into full service to help address the biodiversity crisis. The international body for biodiversity is the Intergovernmental Science-Policy Platform on Biodiversity and Ecosystem Services (IPBES). The IPBES recently raised the alarm about the unprecedented and accelerating species extinction rates; among other things it called for enhanced environmental monitoring and evaluation (IPBES, 2019). For computational bioacoustics to address such global challenges fully, we must focus openly and collaboratively on integration work.

Behaviour and multi-agent interactions

Animal behaviour research (ethology) can certainly benefit from automatic detection of vocalisations, for intra- and inter-species vocal interactions and other behaviour. This will increasingly make use of SED/SELD/object-detection to transcribe sound scenes. Prior ethology works have used correlational analysis, Markov models and network analysis, though it is difficult to construct general-purpose data-driven models of vocal sequencing (Kershenbaum et al., 2014; Stowell, Gill & Clayton, 2016a). Deep learning offers the flexibility to model multi-agent (i.e. multi-animal) sound scenes and interactions, with recent work including neural point process models that may offer new tools (Xiao et al., 2019; Chen, Amos & Nickel, 2020b).

Ethology is not the only reason to consider (vocal) behaviour in DL. The modelling just mentioned is analogous to the so-called “language model” that is typically used in automatic speech recognition (ASR) technology: when applied to new sound recordings, it acts as a prior on the temporal structure of sound events, which helps to disambiguate among potential transcriptions (O’Shaughnessy, 2003). This structural prior is missing in most approaches to acoustic detection/classification, which often implies that each sound event is assumed to occur with conditional independence from others. Note that language modelling in ASR considers only one voice. A grand challenge in bioacoustic DL could be to construct DL “language models” that incorporate flexible, open-set, agent-based models of vocal sequences and interactions; and to integrate these with SED/SELD/object-detection methods for sound scene transcription. Note that SED/SELD/object-detection paradigms will also need to be improved: for example the standard approach to SED is not only closed-set, but does not transcribe overlapping sound events within the same category (Stowell & Clayton, 2015). Analogies with natural sound scene parsing may help to design useful approaches (Chait, 2020).

Low impact

When advocating for computational work that might be large-scale or widely deployed, we have a duty to consider the wider impacts of deploying such technology: carbon footprint, and resource usage (e.g. rare earth minerals and e-waste considerations of electronics). Lostanlen et al. (2021b) offer a very good summary of these considerations in bioacoustic monitoring hardware, as well as a novel proposition to develop batteryless bioacoustic devices.

For DL, impacts are incurred while training a NN, and while applying it in practice: their relative significance depends on whether training or inference will be run many times (Henderson et al., 2020). Happily, the power (and thus carbon emission) impacts of training DL can be reduced through some of the techniques that are already in favour for cross-task generalisation: using pretrained networks rather than starting training from random intialisation, and using pretrained embeddings as fixed feature transformations. Data augmentation during training can increase power usage by artificially increasing the training set size, but this can be offset if the required number of training epochs is reduced. The increasing trend of NN architectures designed to have a low number of parameters or calculations (ResNet, MobileNet, EfficientNet) also helps to reduce the power intensity.

The question of running DL algorithms on-device brings further interesting resource tradeoffs. Small devices may be highly efficient, and might neccessarily run small-footprint NNs. (Note that a given algorithm may have differing footprint when run as a Python script on a general-purpose device, vs a low-level implementation for a custom device). Running DL on-device also offers the ability to reduce storage and/or communication overheads, by discarding irrelevant data at an early stage. Alternatively, in many cases it may be more efficient to use fixed recording schedules and analyse data later in batches (Dekkers et al., 2022). The question becomes still more complex when considering networking options such as GSM/LoRa or star- vs mesh-networking.

Our domain has only just begun to spell out these factors coherently. Smart bioacoustic monitoring has potential to provide rapid-response ecosystem monitoring, in support of nature-based solutions in climate change and biodiversity. This motivates further development in low-impact bioacoustic DL paradigms.

Conclusions

In computational bioacoustics, as in other fields, DL has enabled a leap in the performance of automatic systems. Bioacoustics will continue to benefit from wider developments in DL, including methods adapted from image recognition, speech and general audio. It will continue to be driven by the availability of data, developments in audio hardware and processing power, and the demands of national and international biodiversity accounting. However, we cannot respond to these drivers merely by adopting techniques from neighbouring fields. The roadmap presented here identifies topics meriting study within bioacoustics, arising from the specific characteristics of the data and questions we face.

Supplemental Information

Supplemental Information 1 Bibtex database of literature used.

Click here for additional data file.

Additional Information and Declarations

Competing Interests

Author Contributions

Data Availability

Dan Stowell is an Academic Editor for PeerJ.

Dan Stowell conceived and designed the experiments, performed the experiments, analyzed the data, prepared figures and/or tables, authored or reviewed drafts of the paper, and approved the final draft.

The following information was supplied regarding data availability:

This is a review paper and there is no raw data.

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
