# Peer review of "Computational bioacoustics with deep learning: a review and roadmap"

_PeerJ, doi:10.7717/peerj.13152_

## Round 0.1 · original submission · Minor Revisions

The reviewers are generally positive about the manuscript. A revision is needed to fix some remaining issues. Please provide a detailed response letter with your revision. Thanks.

·

Basic reporting

In this manuscript, the author attempts to review the field of computational bioacoustics. To my best knowledge, this is the first comprehensive overview of the field, and I very much appreciate the efforts that went into this report. It is well-written and structured, following the PeerJ guidelines. It is comprehensive and extensive in every aspect and would be a great resource for beginners in the field when published. The scope of the review appears to focus on breadth rather than depth, which I feel was a good choice. The manuscript is well organized and covers all important aspects of computational bioacoustics, sometimes even providing hints and insights into related fields (e.g., deep learning best practices) which is a nice addition. Overall, I feel that this report deserves to be published and would be a great resource for students, competition participants (e.g., LifeCLEF), and computer scientists entering the field.

Experimental design

The search query could have included terms specific to the field like “soundscape”, “recording” or general terms like “identification” or “detection”. Yet, after investigating the reference section, I feel that all major publications in this field have been included and so this might not be a shortcoming after all.

I appreciate the fact that the author limited this review to papers published after 2016 as this year saw in fact significant breakthroughs in the field.

Validity of the findings

The review (by design) is biased towards deep learning, but to be fair, it is the most common way of analyzing large amounts of audio data today and has seen tremendous attention in the past. Therefore, I think this bias is justified.

The author sometimes uses self-references as justification or general reference. Typically, this would not be ideal, but given the author’s previous work and engagement in the field of bioacoustics, it is justified in this context.

I very much like the “Roadmap” section, which is an excellent addition and covers all important areas of future research that I can think of.

Additional comments

The “LifeCLEF2016” reference appears to have a duplicate (2016a and b are the same paper)

·

Basic reporting

Overall, it is a good paper to read reviewing the applications of deep learning on bioacoustic automatic detection, classification, and other works. In general, I wish the paper was better proof-read before its submission. Many acronyms happen for the first time without being spelled out completely. Most of them come out without any short explanations. In addition, there are many small mistakes in the reference, such as in consistent upper or lower cases on months and no year information.

Line 64: We all know NN stands for Neural network. However, people with biology background might not know it immediately. It appears for the first time in the paper here. A full name should be given.

Line 276: On transformer. Facebook presented in 2021 Wav2vec 2.0 for human speech, which was trained self-supervised on sound only and then fine-tuned to other downstream works. Maybe bioacoustics can repeat the success due to the small size of manual labels? I suggest to add one line or two on Wav2vec 2.0 to bring the latest information to the bioacoustics community.

Line 373 occupancy => presence / absence

Line 640: NVIDIA Jetson Nano and Google Coral: should add NVIDIA to be side by side with Google, or remove Google. Either “Jetson Nano and Google Cora” or “NVIDIA Jetson Nano and Google Cora”

Line 650: ARM CMSIS. What is CMSIS abbreviated for? CMSIS appears for the first time in the paper here. A full name should be given.

Line 828: To verify a method’s performance on recognizing individuals needs a data set consisting of labels of individual IDs. So far, there’re not many data sets available since it does need tremendous work of human experts’s effort. However, the role DL plays is not to accomplish the accuracy increase on a dataset but rather, to make prediction on out-of-sample data while keeping the generalization.

Line 847: Detection problem has been well studied in marine mammal / whale community. Many long-term recordings ranging over months to years have few animal calls and need to be detected.

Line 950: What is TDWG?

Line 956: Could you please explain why W3C Provenance Ontology is needed and what benefits it can bring to bioacoustics community?

Line 959: What is “biodiversity crisis” mentioned? Could you please use one sentence to describe what it is?

Line 960: In the term “multi-agent”, does it mean multiple individual animals? I feel confused since this word is usually used in AI literatures for a robot with sensors.

Line 1010: In “It will be continue”, “be” should be removed.

Experimental design

I wish the author can address the small size problem of labelled data. All the problems that deep learning can apply to have been existing for decades and deep learning came as the tool that improves the accuracies on most applications. However, the problem of small data size hinders us from training a model that offers good generalization to new unknown input sounds.

Validity of the findings

No comment

---

## Round 0.2 · accepted · Accept

The reviewers' comments have been addressed. I recommend the publication of the paper.